# Proposal of a Comprehensive and Multi-Component Approach to Promote Physical Activity among Japanese Office Workers: A Qualitative Focus Group Interview Study

**DOI:** 10.3390/ijerph19042172

**Published:** 2022-02-15

**Authors:** Jihoon Kim, Ryoko Mizushima, Kotaro Nishida, Masahiro Morimoto, Yoshio Nakata

**Affiliations:** 1Graduate School of Comprehensive Human Sciences, University of Tsukuba, 1-1-1 Tennodai, Tsukuba 305-8574, Japan; s2130457@s.tsukuba.ac.jp; 2Department of Sports Research, Japan Institute of Sports Sciences, Tokyo 107-0061, Japan; ryoko.mizushima@jpnsport.go.jp; 3Risk Management Dept. 4th, MS &AD InterRisk Research & Consulting, Inc., WATERRAS ANNEX (10F & 11F), 2-105, Kanda Awajicho, Chiyoda-ku, Tokyo 101-0063, Japan; kotaro.nishida@ms-ad-hd.com (K.N.); masahiro-morimoto@ms-ad-hd.com (M.M.); 4Faculty of Health and Sport Sciences, University of Tsukuba, 1-1-1 Tennodai, Tsukuba 305-8574, Japan

**Keywords:** exercise, workplace, health promotion, behavioral research, Japan, social-ecological model, COM-B model

## Abstract

Office workers spend up to two-thirds of their working hours sitting and are less physically active than other occupational groups. It is necessary to develop an effective approach to promote physical activity among office workers. We conducted a focus group interview with seven Japanese office workers to investigate the current status (topic 1) of, and their opinions on (topic 2), physical activity promotion programs in their workplace. Data were analyzed using qualitative data analysis. We classified the data from topic 1 into individual, socio-cultural, physical, and organizational environments. Most participants indicated that they spent a lot of time sitting and needed programs for a wide range of corresponding employee demands. We classified the data from topic 2 into capability, opportunity, and motivation. Most participants indicated that they wanted evidence-based information, a standing desk, and a conducive workplace environment to enhance physical activity. Thus, we proposed a comprehensive and multi-component approach comprising individual (information delivery), socio-cultural environment (team building, supportive atmosphere), physical environment (standing desk, use of poster), and organizational (incentive, encouraging message from an executive, workplace policy) strategies. Future studies should evaluate the effectiveness of this proposal.

## 1. Introduction

Regular physical activity can reduce the risk of mortality and non-communicable diseases, such as cardiovascular disease, cancer, and diabetes [1]. To achieve health benefits, the World Health Organization recommends that adults undertake at least 150 min of moderate-intensity physical activity weekly [2]. Furthermore, the Japanese Ministry of Health, Labour, and Welfare recommends moderate- or vigorous-intensity physical activity for 60 min every day, regardless of any types of physical activity [3]. However, a recent study reported that the prevalence of adults with insufficient physical activity levels was 27.5% worldwide and 30–39.9% in Japan [4]. Moreover, the 2019 National Health and Nutrition Survey conducted in Japan revealed that the age-adjusted average number of daily steps in adults has not increased in the past ten years (7162 steps for men and 6105 steps for women) [5]. Thus, promoting physical activity in the adult population remains a challenge both globally and in Japan.

The major reasons for physical inactivity in modern adults include increased use of motor vehicles, lack of exercise habits in leisure time, and increased sedentary behavior both at work and home [6,7]. Particularly, in the past few decades, physically inactive and sedentary occupations (i.e., prolonged sitting for most of the working hours) have increased owing to the development of science and technology [7]. These working environment changes are considered to be the main causes of physical inactivity and sedentary behavior in adults. Notably, occupational sitting of full-time office employees was associated with a higher risk of diabetes and mortality [8]. Office workers spend more than 70% of their working hours sitting and have the least number of daily steps (6857 steps) compared with other workers, including postal delivery workers (16,100 steps), laborers (12,796 steps), teachers (10,043 steps), academics (9062 steps), and healthcare workers (8072 steps) [9]. Therefore, countermeasures to improve physical activity are necessary for office workers. The workplace may be a crucial setting for this purpose where full-time office employees spend half of their waking time [10]. Numerous recent studies have revealed that workplace-based physical activity promotion interventions have positive effects on physical activity, sitting time, and cardiometabolic markers [11,12,13,14]. Various intervention programs have been adopted such as counseling, goal setting, self-monitoring, the use of print materials and pedometers, incentives (competitions), and environmental development interventions [15]. Furthermore, multi-component interventions are more desirable and effective in increasing physical activity and reducing sitting time than single-component or environmental interventions [11,16].

Despite the benefits of workplace physical activity intervention, many workplaces have difficulty introducing it. According to previous studies, there are still barriers to introducing physical activity promotion in the workplace for various socio-ecological reasons, including individual, environmental, and organizational reasons [17,18,19]. Furthermore, employers and employees have different perspectives to physical activity promotion in the workplace [20,21]. Therefore, in developing workplace physical activity promotion, it is crucial to hear the employers’ and employees’ demands, opinions, and perspectives. Recently, Van Kasteren et al. [22] proposed a hypothesis, which integrated the socio-ecological model (SEM) [23] and the capability, opportunity, motivation, and behavior (COM-B) model [24] to develop an office-based physical activity promotion program. In their proposal, they first classified each intervention component based on SEM factors, such as individual, environmental, and organizational factors. Thereafter, they classified each intervention component based on three essential conditions: motivation, opportunity, and capability. Finally, they emphasized that SEM and COM-B factors should be balanced for systematic physical activity promotion programs. Although Van Kasteren et al.’s hypothesis [22] may be useful for understanding stakeholders’ demands and developing systematic approach, no study used this hypothesis for developing workplace physical activity promotion programs to our knowledge.

In recent years, various policies have been implemented to address labor shortages caused by the declining birthrate and aging population in Japan. The concept of “Kenko Keiei”, which refers to health and productivity management, has attracted increasing attention [25]. Based on this concept, improvements in worker health can increase worker productivity. Since 2017, this concept has been adopted as a policy that has implemented certification and qualification systems to target companies by the Ministry of Economy, Trade, and Industry. According to this policy, certified and qualified companies should be satisfied with a new standard that improves employees’ health by offering various programs. Thus, allowing physical activity promotion approaches in the workplace is as one of the most attractive programs because of its cost-effectiveness [26]. However, there is a lack of research on effective approaches in Japan. Furthermore, because of the differences between Western and Japanese cultures, as well as labor environment and policies, effective approaches to improve physical activity of office workers should be suitable in the Japanese context. To address this gap, we aimed to develop a comprehensive and multi-component approach to promote physical activity among Japanese office workers based on focus group interview (FGI) data.

## 2. Materials and Methods

### 2.1. Study Design and Recruitments

This study adopted a qualitative research design using FGI. FGI is a useful qualitative research method that can understand the perspectives of participants through discussion [27]. Moreover, it has been used to develop health education programs for both individual interventions and population approaches, such as workplace-based interventions [27,28,29]. Participants were recruited from an insurance company in Tokyo through interoffice advertisements. Based on a previous study, which recommended a group of six to twelve participants for FGIs [27], we recruited seven participants (five regular workers, two managers). Participants were eligible if they were 18–64 years old and regular workers or managers. This study was approved by the University of Tsukuba Faculty of Health and Sport Sciences Ethics Committee (approval number: Tai 019-137).

### 2.2. Focus Group Interview and Survey

An FGI was conducted on 22 June 2020 in a meeting room located in the collaborative insurance company in Tokyo. On the day of the interview, there were only 29 COVID-19 cases in Tokyo, and the state of emergency was not declared. Therefore, face-to-face interviews could only be conducted while following precautions such as washing hands, checking body temperature, and wearing masks. However, one participant (regular worker) reported sick (chills) on the interview day; thus, she was interviewed remotely using Zoom Video Communication Inc. (Zoom). Considering the heterogeneity between regular workers and managers in Japanese work culture, two FGI sessions were conducted: one for regular workers (*n* = 5) and one for the managers (*n* = 2). The interview session of the regular workers lasted 80 min, whereas that of the managers lasted 40 min. The interviews were recorded using digital voice recorders (ICD-TX650, Sony) and were transcribed into text. Before conducting the FGIs, the researchers (Y.N. and R.M.) explained the purpose of the study and obtained informed consent from the participants. Data on participant characteristics, such as sex, age, height, weight, job position, education levels, continuous years of employment, and self-reported physical activity (using a Global Physical Activity Questionnaire) [30], were collected. We calculated body mass index (BMI) as weight (kg) divided by squared height (m^2^). Two topics were discussed in the FGI: (1) perception of workers regarding the importance of physical activity and the current status of physical activity promotion programs available in the workplace and (2) the facilitators of, needs for, feasibilities of, and barriers to physical activity promotion interventions. The interview questions asked were as follows: 1-1: How do you perceive the importance of physical activity?; 1-2: How do you recognize the current situation of physical activity in the workplace?; 2-1: What are the useful and demanding factors for promoting physical activity in the workplace?; 2-2: Is it possible to conduct the aforementioned factors in the future? What are the expected barriers?

### 2.3. Analysis

The FGI data were qualitatively analyzed using NVivo software (Release 1.0). A framework method was adopted that can use inductive analysis, deductive analysis, or both using a pre-existing theory [31]. In FGI analysis, Van Kasteren et al.’s hypotheses [22] were used. Topic 1 identifies worker perceptions on the importance of physical activity and the current status of physical activity promotion. We used SEM to analyze topic 1 data. According to SEM, human health behavior is determined by the interaction between the individual and the environment, including social and natural environments and policies [22,23]. Some workplace-based physical activity interventions for office workers have introduced the SEM model to develop intervention programs and reported the effectiveness of SEM-based intervention programs on sitting time and physical activity [11,12]. Therefore, when considering the interaction between daily physical activity and sitting time at work among office workers, it is appropriate to analyze topic 1 data using the SEM model.

Topic 2 aimed to identify the facilitators of, needs for, feasibilities of, and barriers to physical activity promotion interventions. Thus, the COM-B model of behavior was used to organize the participants’ opinions more systematically. This model identifies the following three factors that need to be present for any behavior to occur: capability, opportunity, and motivation [22,23]. Capability is defined as “the individual’s psychological and physical capacity to engage in the activity concerned”. Opportunity is defined as “all those brain processes that energize and direct behavior, not just goals and conscious decision-making”. Lastly, motivation refers to “all the factors that lie outside the individual that makes the behavior possible or prompt it”.

Similar to SEM, the COM-B model is usually used to develop physical activity promotion interventions [22,29,32]. Thus, we deemed it appropriate for analyzing topic 2. The researchers (J.K. and R.M.) independently performed open coding, grouped codes, and created categories. They subsequently discussed the overlapping and non-overlapping contents in the initial analysis. New codes, such as work–life balance, commute, and COVID-19, were added, in addition to Van Kasteren et al.’s [22] hypotheses. Finally, the results were confirmed by three researchers (J.K., R.M., and Y.N.), and a comprehensive and multi-component program based on FGI data was developed.

## 3. Results

### 3.1. Participant Characteristics

The characteristics of the FGI participants are listed in Table 1.

Five office workers were regular workers (two women and three men) and two were managers (2 men). The age and continuous years of employment of the participants ranged from 39 to 62 years and 2 to 39 years, respectively. All participants had undergraduate degrees, suggesting high education levels. The range of values for BMI, overall moderate to vigorous physical activity (MVPA), occupational MVPA, transport-related MVPA, leisure-time MVPA, and sitting time were 20.2–28.7 kg/m^2^, 180–1080 min/week, 0–50 min/week, 50–420 min/week, 0–660 min/week, and 300–720 min/day, respectively.

### 3.2. Topic 1 Analysis

We first classified the data from topic 1 into individual factors, socio-cultural environment factors, physical environment factors, and organizational factors. Further, individual factors were classified into the following three categories: demographic, biological, and psychological. The socio-cultural environment factor was classified into social networks, whereas physical environment factors were classified into the natural environment and built environment. Lastly, organizational factors were classified into policy and organizational culture. Some participants indicated that they were too sedentary and wanted various programs corresponding to employees’ demands. Moreover, there were many comments that physical activity decreased because of the COVID-19 pandemic and the subsequent shift to remote work, and that the environment in the workplace and at home affected physical activity. Furthermore, it was noted that there were more opportunities for family members to exercise together than for colleagues at work and there was little awareness on the importance of physical activity in the workplace (Table 2).

#### 3.2.1. Individual Factors

We classified individual factors into demographic, biological, and psychological factors. The categories demographic and biological were subdivided into codes, such as age and biological health, respectively. Psychological factors were subdivided into codes, including stress, personality, habit, attitude, beliefs, knowledge, experience, and preference. In the demographic category, participants indicated that they felt the importance of physical activity while aging.


*G: As I get older, I start to feel that I have various physical problems. I think it is important to maintain sufficient physical activity to extend my life expectancy as much as possible. (Age)*


In the biological category, participants indicated that biological health affects the perception of the importance of physical activity.


*A: I do not want to move my body because I have a chronic disease. I do not want to be forced to do physical activities. (Biological health)*

*C: I feel the importance of moving my body during remote work because it is too sedentary and leads to pain in the back and shoulder. However, I feel fine when commuting to work, so I think physical activity is important. (Biological health)*


In the psychological category, participants recognized that the following factors affected their perception of the importance of physical activity and the current status of physical activity: stress, personality, habit, attitude, beliefs, knowledge, experience, and preference.


*F: Rather than going out (exercising) to release stress, it is more often the case that we go to drink to complain each other. (Stress)*

*D: It’s not in my nature to sit for a long time, so I often go to the print room to take a printout when sitting for over 1 h. (Personality)*

*G: I am a restless person, so I am conscious of standing up, stretching my back, or going to the print room to take a printout frequently. (Personality)*

*B: Usually, when I am in the office, I use the stairs as much as possible. Once an hour, I stand up from my seat or walk in the hallway to move my body. (Habit)*

*G: Because I want to keep my body moving, I try to use stairs or walk every day instead of using escalators. (Habit)*

*C: I understand that it is important to improve physical activity, but it is not realistic. (Attitude)*

*E: I walk to lose weight or prepare for a disaster such as an earthquake (walking when the train stops due to a major earthquake), but I do not focus on increasing physical activity for its own sake.*

*D: In my opinion, exercise means running or kicking a ball. When I tried to alight one stop early on the commute for a walk, I did not feel any good changes in my body. In particular, because of not reducing my weight, I think exercise is not effective without high intensity. (Belief)*

*D: I am not familiar with the term “physical activity.” It does not seem to be an alternative to exercise. (Knowledge)*

*E: Since I started yoga, I realized that my back muscles get stiff after long sitting. So, I get up to stretch. (Experience)*

*B: Because I love to move my body, I prefer using the stairs rather than the escalators on my way to work. I also go trekking or exercise on weekends because I enjoy it. (Preference)*


#### 3.2.2. Socio-Cultural Environment Factors

We classified socio-cultural environment factors into social networks. Additionally, the social networks category was subdivided into codes including family, peers, and colleagues. In the social networks category, participants indicated that family members have more opportunities to exercise together than work colleagues, and there was little awareness on the importance of physical activity among colleagues.


*C: I live with my husband. When the first state of emergency was announced, we took a walk along the river, which is located around my home. When he was lazy, I invited him to take a walk with me. If he was working out, I thought I had to do it too, so I did it with him. When my husband does something, it stimulates me. (Family)*

*B: In my family, I am the most physically active. Sometimes, I propose that my wife walk, but she is lazy and does not want to do it. So, it is hard to promote physical activity to her.” (Family)*

*G: When my child was young, I used to do a catch ball. I still take a walk with my wife occasionally. She sometimes recommends cycling when I am lazy. (Family)*

*B: I play soccer with younger people in my town once every two weeks. (Peers)*

*G: We do not exercise with my colleagues, so we are not affected by each other. Sometimes, there is a conversation about healthy behaviors such as alighting one stop early on the commute for a walk. (Colleagues)*

*F: Our most colleagues are consultants. We almost always work alone than work as a team, so we do not have the opportunity to exercise together. (Colleagues)*


#### 3.2.3. Physical Environment Factors

We classified the physical environment factors into natural environment and built environment. The natural environment category was subdivided into codes, including COVID-19 and the weather. Built environment was subdivided into codes, including work environment, the environment around the workplace, residential environment, and commuting. In the natural environment category, participants indicated that the prevalence of COVID-19 leads to physical inactivity and that the weather seems to be related to physical activity.


*G: COVID 19 deprived us of the opportunity to exercise with our family. (COVID-19)*

*F: Now I cannot go to the gym because of COVID-19. I felt well when I moved my body. (COVID-19)*

*E: I used to walk 200,000 steps a month, but in April and May (the state of emergency), I only walked 70,000 steps. I felt that daily physical activities such as commuting are crucial. (COVID-19)*

*A: The river flows from my house to the nearest station. When cherry blossoms bloom, I take a walk with my wife. Just during this season, I become active. (Weather)*


In the built environment category, participants indicated that work environment, the environment around the workplace, residential environment, and commuting can affect their physical activity and sedentary behavior.


*F: In my case, I have a lot of desk-based work. (Work environment)*

*G: My job is desk-based. Because I sit down for a long time, my back hurts a little. (Work environment)*

*A: I do not have any interest in walking around my house or workplace. (Environment around the workplace)*

*C: There is a green space around the office. I feel grateful because I can see seasonal flower blooms when I commute or go out. (Environment around the workplace)*

*D: There is a green space around the office, but the roads are messy. In my first days at this company, I tried to run around the office; however, I was stopped by traffic lights many times. There were a lot of people around the worksite, and I thought it was difficult to run. (Environment around the workplace)*

*B: I moved to the suburbs 30 years ago because l like the sea and green space and wanted to get some rest on the weekend. There are many good places to walk near my house. (Residential environment)*

*C: There is a waterfowl habitat near my house (a five-minute walk), where pheasants live and I can hear their yelling. Furthermore, the sidewalks are wide, and the roads are well-maintained. I think it is a very good running environment. (Residential environment)*

*F: The sidewalk around my house is quite narrow. Therefore, it was difficult to walk along the sidewalk when my kids are little because of many vehicles. (Residential environment)*

*A: My main stress was commuting, which takes an hour and a half one-way. However, after the shift to remote work during the COVID-19 pandemic, this stress is relieved. I think reducing commuting time is more crucial for my health than doing exercise or reducing sitting time. (Commuting)*


#### 3.2.4. Organizational Factors

We classified organizational factors into policy and organizational culture categories. These categories were subdivided into codes, including policy and work–life balance, respectively. In the policy category, participants indicated that they wanted various programs corresponding to employee demands.


*B: I think that our company has introduced various health promotion programs compared with other companies. However, it seems that only a few employees participated in these programs. In addition to physical activity promotion, there may be various programs, such as yoga and sports. (Policy)*

*A: I think the current status is nice, because I just have to choose from various health promotion programs. If the company forces us to do something, I will strongly oppose. (Policy)*

*D: I know there are many health promotion programs introduced by the company, but I do not participate in these programs. (Policy)*


In the organizational culture category, participants indicated that work–life balance may affect participation in health promotion programs in the workplace.


*E: I am so busy with my work, and it is a priority for me. Therefore, I have not participated in any health promotion programs. (Work–life balance)*

*C: In the workplace, I think that I must focus on my work rather than spend time on myself, which participation in health promotion programs entails. (Work–life balance)*


### 3.3. Topic 2 Analysis

We classified the data from topic 2 into the following factors: capability, opportunity, and motivation based on the COM-B model. The capability factor was further classified into psychological capability. The opportunity factor was classified into physical and social opportunities. The motivation factor was classified into automatic and reflective motivations. Some participants indicated that they wanted evidence-based information; various incentives; and a conducive work environment for promoting physical activity, such as a standing desk or shower room (Table 3).

#### 3.3.1. Capability Factors

The capability factors included a psychological category that was subdivided into education. In this category, participants indicated that they wanted evidence-based information.


*A: I want evidence-based information associated with health. Furthermore, when offering information, it may be helpful to explain the mechanism of health benefits to the employee because they already have basic knowledge on health.” (Education)*


#### 3.3.2. Opportunity Factors

We classified opportunity factors into physical and social opportunities. The physical opportunity category was subdivided into codes, including environmental restructuring and enablement. Social opportunity was subdivided into codes including environmental restructuring and restriction. In the physical opportunity category, participants indicated that they needed physical environmental restructurings, such as a shower room and a movable sit/standing desk.


*D: I think it is good to install a shower room or locker in this building. When I walk to the office after getting off one station earlier, I am drenched in sweat. (Environmental restructuring)*

*E: If physical activity can be improved by simply shifting from sitting to standing while working, a conference room with a standing table or a movable sit/stand desk could be useful. (Environmental restructuring)*

*G: It will be effective to set some constraints, such as environmental restructuring on the company in the medium term. (Enablement)*


In the social opportunity category, participants raised the need for social environment restructuring, such as creating a culture that permits movement freely in the workplace, and for restrictions, such as encouraging messages (close to command) from an executive.


*A: Our division head is not an active person but is interested in new information. Our group leader takes initiative and acts. Therefore, our team is very active and motivated. (Environmental restructuring)*

*B: I think that it is necessary to create an office culture that allows employees to move without hesitation such as moving their shoulders in circles and standing up. (Environmental restructuring)*

*E: There is nothing better than encouraging messages (close to command) from an executive. However, it is also difficult to establish an office culture. (Restriction)*


#### 3.3.3. Motivation Factors

We classified motivation factors into reflective and automatic motivations. Reflective motivation factors were further subdivided into codes including persuasion, incentivization, and coercion. Automatic motivation factors were subdivided into codes, including incentivization and modelling. In the reflective motivation category, participants cited that the use of posters, incentives, and policies may be effective strategies to improve physical activity.


*C: I think poster pop, which is attached near the printer, is useful to promote physical activity during the use of the printer. (Persuasion)*

*F: The recommendation from the superior may be effective. Like “Let’s take a break or reduce overtime work”, it would be nice to declare “Let’s make a move.” (Persuasion)*

*E: Incentives may be a very effective strategy for promoting physical activity with immediate and sustainable effects. We may be satisfied if we can select an incentive as we want from many types, such as money, gift certificates, book cards, or points. (Incentivization)*

*F: Working reform, which is reducing overtime work by switching off their computer and office if the specified time is exceeded, is progressing. Although there are many opinions on this, we will get used to the company policy. (Coercion)*


In the automatic motivation category, participants suggested that various incentives and policies (group-based competition or employees’ performance assessment) may be effective to improve physical activity.


*C: Incentive compensation by interlocking their daily step count may be effective. However, if they do not receive the incentives they want, they may quit. Sometimes, they may find being physically active to be fun. (Incentivization)*

*C: If the amount of physical activity is incorporated into the employees’ performance assessment, I think all employees will participate in improving physical activity. But I think it is a big no-no by Japanese culture. (Modelling)*

*B: If we aim to improve the level of physical activity at our company, why do we not give incentives to groups that achieve sufficient physical activity by group-based competition? (Modelling)*


### 3.4. Comprehensive and Multi-Component Intervention Program

Based on the FGI results, we developed a comprehensive and multi-component intervention program comprising individual (information delivery), sociocultural environment (team building, supportive atmosphere), physical environment (standing desk, use of posters), and organizational strategies (incentives, encouraging message from an executive, workplace policy) (Figure 1).

## 4. Discussion

### 4.1. Principal Findings

In this study, an FGI was conducted to develop a comprehensive and multi-component approach to promote physical activity among Japanese office workers. The FGI explored (1) perceptions of the importance of physical activity and the current status of physical activity promotion and (2) the facilitators of, needs for, feasibilities of, and barriers to physical activity promotion interventions.

From the FGI data of topic 1, the results showed that Japanese office workers’ perceptions of the importance of physical activity and the current status of physical activity promotion are affected by socio-ecological factors, such as individual, socio-cultural environment, physical environment, and organizational factors. Among the individual factors, age, biological health, and psychological state were associated with the perception and current status of physical activity [33,34,35]. In particular, psychological state was perceived to be strongly associated with the perception of the importance of physical activity and current status. Socio-cultural environment factors that may affect physical activity were also identified, including family, peers, and colleagues [36,37]. Family was perceived to be strongly associated with physical activity. Although the FGI was conducted in an occupational setting, there were only a few comments regarding work colleagues. There seemed to be few intercommunications with each other about physical activity in the workplace. In this regard, we propose that team-building activities that promote intercommunication will be an effective approach to promote physical activity in the workplace. Physical environment factors were also identified, such as the COVID-19 pandemic, the weather, work environment, the environment around the workplace, residential environment, and commuting, which may affect the perception and current status of physical activity [38,39,40,41]. In particular, COVID-19 and the environment around the workplace were perceived to be strongly associated with physical activity levels. Due to the COVID-19 pandemic, people could not experience their daily normal life activities, such as commuting and going to the gym. There seemed to be a need for a new normal approach, i.e., non–face-to-face and not a conventional approach, to promote physical activity in office workers. Similar to previous studies [40], the environment around the workplace was perceived as a motivation for workers to engage in physical activity. Furthermore, organizational factors, such as policy and work–life balance, were identified; they may be associated with the perception and current status of physical activity [42,43]. In particular, policy in the workplace was perceived to be strongly associated with physical activity. Workers seemed satisfied with the current company policies but wanted various health promotion programs from which they could choose. Therefore, offering multi-component physical activity promotion programs may be an effective approach.

From the FGI data of topic 2, the results showed many facilitators of, needs for, feasibilities of, and barriers to physical activity promotion in Japanese office workers. Regarding capability factors, evidence-based information included education, which may be an effective component for this occupational group. According to a previous study [15,44], delivering information is one of the most adopted workplace physical activity programs which can be effective. Additionally, adopting such a program is cost-effective. Therefore, delivering evidence-based information is a component that can be introduced in the future.

Regarding opportunity factors, physical and social environmental changes were perceived as effective components to offer opportunities to promote physical activity. The participants mentioned that installing a standing desk or shower room could promote physical activity and create an office culture that allows employees to be physically active without hesitation. Physical and environmental renovation can be effective [11,12,13,14,15] for workplace-based interventions. In particular, standing desks have been reported to be effective in reducing sitting time and the risk of cardiovascular biomarkers among office workers [12,13,14]. Without considering their cost, it would be the most effective intervention to implement. Moreover, social-environmental changes, such as creating an office culture and encouraging messages from an executive, could also be introduced in the future because of their low cost.

Regarding motivation factors, posters, various incentives, policies, and team-building activities (group-based competition) were identified. According to a previous study, posters and team-building activities were among the most frequently implemented workplace physical activity programs and can be feasibly introduced because of their low cost [15,45]. Moreover, incentivization is considered the most effective component in promoting physical activity [46,47]. However, implementing various incentives and policies is challenging because of the associated high cost and revision of the administrative procedures of the company.

Based on the FGI results, a comprehensive and multicomponent approach (Figure 1) was developed. Information delivery was selected as an individual strategy and considered as a capability factor that can improve and offer knowledge and skills [22,23,24]. As a socio-cultural environment strategy, a team-building activity and a supportive atmosphere were proposed. Team building can stimulate emotional responses, whereas a supportive atmosphere is an opportunity factor that can improve physical activity levels [22,23,24]. As a physical environment strategy, standing desks and posters were selected. Standing desks were regarded as an opportunity factor that can make physical activity, such as standing, possible. The use of posters is a motivation factor that can stimulate emotional response or analytical decision making [22,23,24]. Regarding organizational strategies, we proposed incentivization and encouraging messages from an executive to stimulate emotional responses. Moreover, workplace policy was adopted as an opportunity factor that can prompt physical activity [22,23,24]. A future study will evaluate the feasibility and efficacy of the comprehensive and multicomponent intervention program.

### 4.2. Strengths and Limitations

The strength of this study is that it investigated the perceptions and current status using FGI to develop a comprehensive and multi-component physical activity promotion program that was suitable for Japanese workplace culture and working environment. To the best of our knowledge, few studies have explored this topic using a qualitative research method in Japan. Furthermore, FGI data were analyzed using Van Kasteren et al.’s hypotheses [22], which integrated the social-ecological model and the COM-B model [22,23,24]. To develop physical activity promotion programs for office workers, the workers emphasized that it is crucial to understand their occupational nature and daily life from a social-ecological perspective, and that intervention programs associated with both motivation as well as capability and opportunity are needed. This study demonstrated that Van Kasteren et al.’s [22] hypotheses can be used for FGI data analysis and to develop physical activity programs.

This study had several limitations. First, the FGI was conducted with a small number of participants from one office-based insurance company. Although we attempted to avoid bias in the FGI group by considering the educational background, sex, and job position (regular worker, manager), our results may still be biased because of the small sample size and the limited recruitment process. Therefore, it is necessary to conduct FGIs involving workers of other desk-based work companies, while considering various age groups, sex, and a sufficient sample size. Second, one participant reported sick (slight cold) on the day of the interview, so the interview was conducted remotely. According to the FGI guidelines [48], the FGI should be conducted in an environment comfortable for participants. Although the interviewers attempted to cultivate a comfortable atmosphere where the participants can speak freely, even in remote settings, the FGI of this particular participant was not administered under ideal conditions. Third, in a previous study [4], about 30–39.9% of Japanese adults have insufficient physical activity levels. However, all participants of the present study met the WHO physical activity recommendations (at least 150 min of MVPA weekly) [2]. This could limit the generalizability of the present study. Fourth, we focused only on office workers. It is necessary to carefully adopt the current proposal at other worksites where workers have high physical workloads. In addition, developing the program for high physical workload workers may be challenging. Fifth, the present study is a proposal and has not yet identified the effectiveness. Therefore, a future study will test the effectiveness of this approach. Finally, there was an issue regarding the time allotted to the topics in the FGI. The regular office workers discussed topics 1 and 2 for 55 min and 25 min, respectively, whereas the managers discussed them for 25 min and 15 min, respectively. Considering this difference in time allotment, there might not have been sufficient time to adequately address topic 2. Therefore, future studies should consider balanced time allotments when conducting FGIs.

## 5. Conclusions

This study developed a comprehensive and multi-component approach to promote physical activity in Japanese office workers using FGI data. The proposed comprehensive and multi-component approach comprised individual (information delivery), socio-cultural environmental (team-building activities, supportive atmosphere), physical environmental (standing desk, poster), and organizational (incentive, encouraging messages from an executive, workplace policy) strategies. It is necessary to evaluate the feasibility and effectiveness of our proposed approach in a future study.

## Figures and Tables

**Figure 1 ijerph-19-02172-f001:**
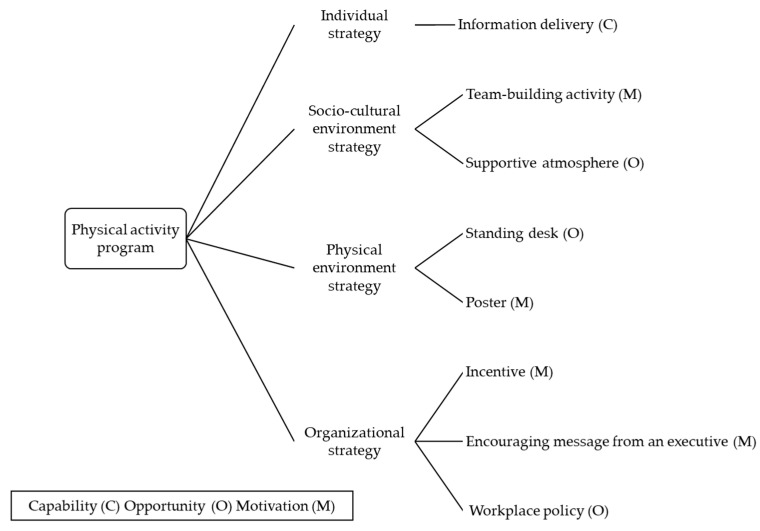
Comprehensive and multi-component approach based on the focus group interview data.

**Table 1 ijerph-19-02172-t001:** Participant characteristics.

Variables	Participants (*n* = 7)						
	A	B	C	D	E	F	G
Job position	RG	RG	RG	RG	RG	MG	MG
Sex	Male	Male	Female	Male	Female	Male	Male
Age (years)	57	62	39	48	42	47	53
BMI (kg/m^2^)	28.0	22.7	21.4	24.6	20.2	28.7	21.1
Education level	UG	UG	UG	UG	UG	UG	UG
Continuous years of service	39	2	10	7	9	23	31
Self-reported physical activity							
Overall MVPA (min/week)	270	1080	365	410	335	210	180
Occupational MVPA (min/week)	0	0	50	0	10	0	0
Transport-related MVPA (min/week)	270	420	165	50	175	150	120
Leisure-time MVPA (min/week)	0	660	150	360	150	60	60
Sitting time (min/day)	720	300	300	540	540	510	600

Note: BMI, body mass index; MVPA, moderate to vigorous physical activity; MG, manager; RG, regular worker; UG, undergraduate degree.

**Table 2 ijerph-19-02172-t002:** Mapped results of topic 1 based on SEM.

Factors	Categories	Codes
Individual	Demographic	Age
	Biological	Biological health
	Psychological	Stress
		Personality
		Habit
		Attitude
		Beliefs
		Knowledge
		Experience
		Preference
Socio-cultural environment	Social networks	Family
		Peers
		Colleagues
Physical environment	Natural environment	COVID-19
		Weather
	Built environment	Work environment
		Environment around workplace
		Residential environment
		Commuting
Organizational	Policy	Policy
	Organizational culture	Work–life balance

Note: SEM, social-ecological model [22,23].

**Table 3 ijerph-19-02172-t003:** Mapped results of topic 2 based on COM-B.

Factors	Categories	Codes
Capability	Psychological	Education
Opportunity	Physical	Environmental restructuring
		Enablement
	Social	Environmental restructuring
		Restriction
Motivation	Reflective	Persuasion
		Incentivization
		Coercion
	Automatic	Incentivization
		Modelling

## Data Availability

The data presented in this study are available on request from the corresponding author. The data are not publicly available due to the protecting privacy.

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
