# Peer review of "Proposal of a Comprehensive and Multi-Component Approach to Promote Physical Activity among Japanese Office Workers: A Qualitative Focus Group Interview Study"

_ijerph, 2022, doi:10.3390/ijerph19042172_

Round 1

Reviewer 1 Report

The paper properly proposed a comprehensive and multi-component intervention program based on analysis on topic 1 and topic 2. 
The article is well structured. However, It needs to provide the theoretical background of the study model. 
I would like to recommend for authors to revise the title by referring to the following; 'A suggestion of a Comprehensive and Multi-component Intervention Program...', or 'An analysis of a suggestion of a Comprehensive and Multi-component Intervention Program......'.
In Introduction: It needs to address the research trend in the field referring previous studies.
In the Literature Review: There is currently no literature review. It needs to provide theoretical background of topic 1 and topic 2 factors. 
The methods are adequately described. 
Line 179: It needs to provide a detailed explanation of SEM

Reviewer 2 Report

The current study aimed to develop a comprehensive and multi-component approach to promote physical activity among Japanese office workers based on focus group interview (FGI) data. The study adopted a qualitative research design using FGI. 

The study's subject is interesting. Since health promotion programs should be developed based on participants' demands, FGI seems to be a good approach. However, there are major concerns that must be addressed.

The authors reported some data of steps recommendation per day, which is a important information, otherwise some references of global recommendations of frequency, intensity, type and time are important too (such as FITT principle - see Thompson PD, Arena R, Riebe D, Pescatello LS. ACSM's New Pre participation Health Screening Recommendations from ACSM's Guidelines for Exercise Testing and Prescription, Ninth Edition. Curr Sports Med Rep 2013; 12:215 – 217.)

Although the present study was aimed to developed a  comprehensive and multi-component approach to promote physical activity, it was not able to investigate if this intervention is effective or not. Thus, the development, in my opinion, is concluded since the intervention effectiveness' is tested. It is a major  limitation of the present study?   

Minor comments 

Title [suggestion] - A Comprehensive and Multi-component Approach to Increase Physical Activity Among Japanese Office Workers: A Qualitative Focus Group Interview Study 

Reviewer 3 Report

This paper entitled “A Comprehensive and Multi-component Intervention Program to Increase Physical Activity Among Japanese Office Workers: A Qualitative Focus Group Interview Study”. They proposed the novel a comprehensive and multi-component physical activity promotion suitable for the Japanese workplace culture and working environment. The model was developed based on the integration of the social-ecological model and the COM-B model which is cover all perspectives of model developing. However, I have some suggestion to improve your manuscript as the following:

  1. In title, I suggest to change from “..to increase physical activity…” to “to promote physical activity” which corresponded to the objective of this study.
  2. I also confused about the unit of PA measured by the GPAQ (in LINE 159-163) that are not same as the unit in the table 1. Based on calculation and analysis of GPAQ, PA at work, transportation and leisure were calculated and presented in MET-min/week. And there are no guideline or constant MET value to calculate the sitting time in MET-min/week as shown in table1. Which is formula used?
  3. I suggest to add the limitation of this study. All subjects are low active (overall PA<600 MET-min/week) that might affect the conclusion in the model. And suggest to develop a focus group interview in high active workers in the future study.

Thank you

Round 2

Reviewer 2 Report

My all concerns were achieved.